# The impact of sampling bias on viral phylogeographic reconstruction

**Pengyu Liu** [ID], **Yexuan Song, Caroline Colijn, Ailene MacPherson** [ID] *

Department of Mathematics, Simon Fraser University, Burnaby, British Columbia, Canada

* ailenem@sfu.ca

## Abstract

Genomic epidemiology plays an ever-increasing role in our understanding of and response to the spread of infectious pathogens. Phylogeography, the reconstruction of the historical location and movement of pathogens from the evolutionary relationships among sampled pathogen sequences, can inform policy decisions related to viral movement among jurisdictions. However, phylogeographic reconstruction is impacted by the fact that the sampling and virus sequencing policies differ among jurisdictions, and these differences can cause bias in phylogeographic reconstructions. Here we assess the potential impacts of geographic-based sampling bias on estimated viral locations in the past, and on whether key viral movements can be detected. We quantify the effect of bias using simulated phylogenies with known geographic histories, and determine the impact of the biased sampling and of the underlying migration rate on the accuracy of estimated past viral locations. We find that overall, the accuracy of phylogeographic reconstruction is high, particularly when the migration rate is low. However, results depend on sampling, and sampling bias can have a large impact on the numbers and nature of estimated migration events. We apply these insights to the geographic spread of Ebolavirus in the 2014-2016 West Africa epidemic. This work highlights how sampling policy can both impact geographic inference and be optimized to best ensure the accuracy of specific features of geographic spread.

## Introduction

Genomic epidemiology refers to the use of pathogen genomes to understand how infectious diseases are transmitted across a range of scales, from local outbreaks to global geographic spread. Due to improvements in sequencing technology and decreasing cost, genomic epidemiology has increased greatly in scale in many jurisdictions and for many viruses to the extant that sequencing now plays an important role in pathogen surveillance [1]. Sequencing allows us to identify new viruses, understand their origins and natural reservoirs, to characterize their transmission dynamics in human populations, and to understand their evolution [2, 3]. Interest in genomic surveillance for viruses has grown, in the context of the high utility of genomic data for these tasks and the increasing awareness of the severe threat that emerging viruses can represent for global public health. At the time of writing, the Global initiative on sharing all influenza data (GISAID) repository [4, 5] has over 10 million SARS-CoV-2 sequences and

**Data Availability Statement:** Code and data for simulations and analyses conducted in this paper are available at https://github.com/pliumath/sampling-bias.

**Funding:** C.C. is supported by the Government of Canada's Canada 150 Research Chair program. A. M. is supported by a Natural Sciences and Engineering Research Council of Canada grant (RGPIN-2022-03113). The funders had no role in study design, data collection and analysis, decision to publish, or preparation of the manuscript.

**Competing interests:** The authors have declared that no competing interests exist.

over 350,000 influenza virus sequences. Nearly 3000 Zaire Ebolavirus sequences are accessible through ViruSurf along with over 12,000 Dengue virus sequences and sequences of MERS, other Ebolaviruses and Severe Acute Respiratory Syndrome coronavirus 1 (SARS-CoV-1).

Virus sequences are a rich potential source of information about the origins, past population dynamics and geographic movements of viruses [6]. This is in part because sequences reveal information about viral evolution and population dynamics at times and in places where the viruses were not directly observed. Sequences are interpreted with the aid of phylogenetic trees, which represent the evolutionary relationships between a set of taxa. Here we focus on the fact that the geographic locations of the virus' ancestors in the past can be reconstructed by extrapolating the location of the observed sequence back in time from the present day [7–11]. There are a number of methods that perform this extrapolation. These treat the geographic location as either a discrete or continuous trait evolving on the phylogenetic tree; some methods use Bayesian approaches to simultaneously reconstruct the location and the phylogeny and others reconstruct the geographic location on a fixed phylogeny [12, 13].

Indeed, phylogeography is one of the main applications of large-scale virus sequence datasets. In 2007, phylogeographic analysis revealed Guangdong as a likely source of Avian H5N1 influenza viruses, and Indonesia a likely sink [9]. Phylogeographic analysis was used to understand transmission dynamics of Ebolavirus in space and time in the Democratic Republic of Congo in 2018–2020, and was part of the genomic surveillance system informing the response to Ebolavirus in real time [14]. In human influenza, phylogeographic analysis is used to characterize the emergence and global spread of human seasonal influenza viruses and compare their circulation patterns [15]. In the SARS-CoV-2 pandemic, phylogeography has been used at a range of scales, to uncover the virus' early dissemination from Europe in 2020 [16], to demonstrate repeated introductions and localized spread in Spain [17], Peru [18], Canada [19] and other settings, and to visualize geographic movements (e.g. in nextstrain) [20].

A number of authors have noted that phylogeographic reconstructions depend on the fraction of infections that are sampled, sequenced and shared, and that phylogeographic estimates can be biased as a result [15, 21–24]. These authors have taken different approaches to compensating for this bias. de Maio *et al.* [23] propose a method called the BAyesian STructured coalescent Approximation (BASTA) to approximate the structured coalescent, having found that discrete trait analysis (DTA), in which migration among locations is treated in the same manner as mutation, gives biased estimates of the migration rates if the locations are strongly non-representatively sampled. Perhaps more fundamentally for viral phylogeography, they also point out that the relative sampling intensities of the locations in DTA-based phylogeography are treated as data and inform the migration estimates. This makes phylogeographic estimates sensitive to sampling choices and can lead to erroneous inferences and erroneously small apparent uncertainties; the structured coalescent approach does not have this issue, and parameter estimates in simulations were better than those from DTA models [23]. Magee and Scotch characterize the impact of two sampling schemes and a range of generalized linear models on the estimation of the root state (point of origin) and on the variables (in the linear model) that are deemed to be important [25]. Bias can occur both in estimates of the parameters of the process (e.g. migration or dispersal rates) and in the reconstructed ancestral geographic locations themselves. Approaches to adjust for bias are in development. In the context of the structured coalescent model, De Maio et al. account for the effect of sampling bias on the estimation of migration rates by integrating over all possible migration histories [23]. Guindon and De Maio address non-uniform sampling in a diffusion model of movement in continuous space by using a Bayesian approach to include the sampling process in the posterior distribution, but this requires an exchange algorithm that introduces substantial computational complexity [26]. Their correction leads to differences in the inferred growth rate,

effective population size, dispersal parameter and the time of the most recent common ancestor (compared to not correcting for the sampling process). Kalkauskas *et al.* explore adding "empty" viral sequences in a continuous-space model [24], finding that this approach can bias in estimates of the root location, and diffusion rate, but not eliminate it. Lemey et al. incorporate both "empty" viral sequences and individual travel history in part to overcome sampling bias [8] in phylogeography. Indeed, the role of travellers is of course important in shaping viral geographic movements themselves, but also potentially important in reconstructing them. In some jurisdictions, travellers' samples are prioritized for sequencing as part of monitoring viral diversity [27]. The information about which sequences are from traveller samples is not typically shared to public databases such as GISAID, which could additionally confound phylogeographic inferences.

Here we focus on maximum-likelihood phylogeographic inference with location modeled as a discrete quantity. Discrete space is appropriate for many epidemiological applications where data is both collected and policies implemented at a regional level. While methods have primarily been developed with Bayesian tools, and so have inherited the disadvantage that limited numbers of sequences can be included if the analysis is to be computationally tractable. An additional challenge is that Bayesian phylogenetic reconstruction produces multiple (posterior) phylogenies with distinct internal nodes. Without comparable internal nodes from tree to tree, it would not be a well-posed task to summarize the nodes' locations, and thus to infer the phylogeographic information. In practice, users of phylogeography who have thousands of sequences often cannot use state-of-the-art Bayesian methods due to computational limitations of these methods for large datasets, and instead use a simpler maximum-likelihood approach [19]: construct a phylogeny, and then reconstruct the geographic locations of the past nodes on that fixed phylogeny, both with maximum-likelihood methods [11].

The impact of location-based sampling bias on the phylogeographic reconstructions of the past locations of pathogens has not been thoroughly characterized, in either Bayesian or maximum likelihood methods. Yet this has important practical implications for the users of phylogeography, particularly where large datasets are used to reconstruct a virus' global geographic movement. The implications of sampling bias for the design of genomic surveillance and data sharing strategies are not well understood. Here we characterize the impact of sampling bias on error in the maximum-likelihood phylogeographic reconstruction of a virus' location in the past. We begin by using simulations of viral branching, cure (or host death), and migration to quantify the impact of sampling bias on the overall reconstruction accuracy, on the types of errors that result, and on the sources of variability in the quality of reconstruction. This exploration includes the impact of sampling bias on location of individual ancestral nodes, the inference of the root location (origin), and the impact of over/under representing recent travelers. To illustrate the potential impact of sampling in a natural context, we then perform phylogeographic analysis using Ebolavirus sequences from West Africa. Throughout, we offer practical comments on when geographic sampling bias is likely to impact maximum-likelihood phylogeographic reconstructions.

## Materials and methods

### Assumptions and tree simulation

We simulate pathogen diversification under the binary-state speciation and extinction (BiSSE) model [28] implemented in the R package diversitree [29]. Simulation code is available on *github* at https://github.com/pliumath/sampling-bias. Here we will consider the case where the two states represent distinct geographical locations. Each node of the resulting rooted binary-state phylogenetic tree is in either location A or location B. Without loss of generality, the

diversification is simulated with the initial lineage (the root of the tree) in location A. We assume that speciation and extinction rates are independent of a lineage's location (i.e. location is a neutral character), though we relax this assumption in the Supplement (see S1 Text). While this assumption is unlikely to be true in natural systems, it allows us to focus on geographical differences in sampling intensity in isolation. As with the speciation and extinction rate, we assume that migration is symmetrical between the two locations. The resulting diversification model has three parameters: the speciation rate $\lambda$, the extinction rate $\mu$, and the migration rate $\alpha$ between location A and location B. In the context of a pathogen phylogenetic tree, "extinction" represents the end of the infectious period (which could be death, or recovery, successful treatment, effective quarantine, etc). Similarly, "speciation" is usually assumed to be coincident with pathogen transmission between hosts.

Observed at the present day, a phylogenetic tree generated by this model has two types of tips: "extinct" tips associated with extinction events some of which, importantly for us here, are assumed to be coincident with sampling and sequencing of the pathogen lineage and "extant" tips representing lineages that exist and remain unsequenced at the present day. As we assume that the sampling and sequencing of a pathogen is coincident with treatment or quarantine the result is a heterochronous time tree consisting of a subset of the extinct tips. To obtain this tree, we drop all extant (continuing past the present day, and not sampled, as sampling occurs through time in the past) lineages of the simulated tree. We call the resulting tree consisting of only "extinct" tips the "true tree", because it has all of the tips that could possibly have been sampled. The geographical location of the internal nodes of this (simulated) true tree are known and will be used as the reference for quantifying the accuracy of ancestral state reconstruction. We select $n$ tips from the true tree, where $n$ reflects sequencing capacity. We call the resulting subtree with only the $n$ selected tips a "downsampled tree". In order to ensure that all $n$ sampled lineages may be in either location, the initial simulation for a true tree is terminated only once it contains at least $n$ tips in location A and $n$ tips in location B.

### Downsampling schemes

To assess the impact of location-biased sampling on ancestral state reconstruction we consider two sampling schemes. In the first scheme, we construct a downsampled tree $S_k$ with $n$ tips from a true tree $T$. The fraction of tips in the downsampled tree that are from location A is $k$ (up to rounding, e.g. if $k = 1/3$ and $n = 100$ we have 33 location-A tips in $S_k$. Mathematically: the tips in $S_k$ are uniformly selected at random from the extinct tips of $T$ such that $\lfloor kn \rfloor$ (rounded to the smaller integer of $kn$) tips are in location A and $n - \lfloor kn \rfloor$ tips are in location B. In our experiments with this downsampling scheme, $k$ ranges from 0.05 to 0.95 in increments of 0.05 (5–95%).

To assess the impact of sampling travellers or those with recent travel-related exposure on phylogeographic reconstruction, we implement a downsampling scheme in which lineages with changes in location near the tips are over- or under- represented. Specifically, we classify a tip in the true tree $T$ as a "recent migrant" if the tip has a different state from its parent node. Suppose $T$ has $m_A$ recent migrants in location A and $m_B$ recent migrants in location B. We construct a downsampled tree $S_p$ with $n$ tips, for which we select tips from $T$, attempting to have a fraction $p$ of the location-A tips be "recent migrants". For example, if $n = 100$, $k = 0.4$ and $p = 0.8$, we want 80% (or 32) of the 40 location-A tips in the downsampled tree to be recent migrant tips. There may be fewer than 32 recent migrant tips in the true tree to begin with. In this case we include all of them. If there are more than 32, we include 32 sampled at random. Mathematically: we uniformly select $\lfloor kn \rfloor$ location-A tips at random such that $\min(m_A, \lfloor kpn \rfloor)$ of the tips are recent migrants, and uniformly select $n - \lfloor kn \rfloor$ location-B tips at random such

that $\min(m_B, \lfloor kpn \rfloor)$ of the tips are recent migrants. In our experiments with this downsampling scheme, we set $k$ to be 25%, 50% or 75%, and $p$ ranges from 5% to 95% in increments of 5%.

## Ancestral state reconstruction

To reconstruct the ancestral states at the internal nodes of the downsampled trees, we use the maximum likelihood method first described by Pagel [30] and implemented in the *ace* function in the R package ape [31]. This method reconstructs the states of internal nodes for a fixed unlabelled tree with known tip states, and gives the likelihood that each internal node (including the root) was in each of the two states (locations). This method also estimates the migration rates of the downsampled trees under the assumption of equal migration rates between the two locations.

## Reconstruction accuracy

We use two quantities to assess the quality of our phylogeographic reconstructions: absolute accuracy and relative accuracy. The absolute accuracy is the fraction of internal nodes for which the reconstructed location is correct. However, if most of the tips and most of the internal nodes are from the same location, the absolute accuracy can be high in a trivial way, as the reconstruction can simply estimate that all the nodes were in that location. The relative accuracy accounts for this. It is the improvement in the absolute accuracy, compared to a null model based only on the locations of the tips.

Mathematically, we define absolute accuracy and relative accuracy as follows. Consider a true tree $T$ and its downsampled tree $S$ with an internal node $i$. Let $c_i$ represents the location of internal node $i$ in the true tree $T$, where $c_i = 1$ if $i$ is in location A and $c_i = 0$ if $i$ is in location B, and $\tilde{c}_i$ be the corresponding likelihood of node $i$ being in location A as inferred by the ancestral state reconstruction method. We define the absolute accuracy of node $i$ as $a_i = 1 - |\tilde{c}_i - c_i|$ and the absolute accuracy of the downsampled tree $S$, $a_S$, as the absolute accuracy averaged over all internal nodes in $S$.

To define the relative accuracy of the downsampled tree $S$, we compare the absolute accuracy $a_S$ to a null expectation $e_S$ assuming that the probability that an internal node is in a given state is proportional to the frequency of the two locations at the tips and hence agnostic of phylogenetic structure. Suppose $S$ has $n_A$ location-A tips and $n_B$ location-B tips. We define the probability that an internal node of $S$ is in location A to be $p = n_A/(n_A + n_B)$ and the expected accuracy of node $i$ as $e_i = 1 - |\tilde{c}_i - p|$. The expected accuracy of the downsampled tree $S$, $e_S$ is the expected accuracy averaged over all internal nodes of $S$, and the relative mean reconstruction accuracy of the downsampled tree $S$ is then given by $r_S = a_S - e_S$.

## Key migration events

To understand the impact of sampling on the reconstruction of "key migration events", events that ultimately lead to large outbreaks following introduction, here we define a procedure for selecting such events from a true tree and measures for quantifying the accuracy of their reconstruction. Noting that the root of a true tree is always in location A, we define a key migration event (KME) in a true tree $T$ to be any edge connecting a location-A internal (parent) node to a location-B internal (child) node which subsequently has at least 15 location-B tips as its descendants. These location-B tips are called the "tips of the KME".

To quantify the different mechanisms resulting in inaccurate reconstruction of KMEs, we characterize the state of a KME in a downsampled tree with two measures: the presence of the internal nodes (the parent node and the child node of the KME) and the presence of sufficient

**Table 1. Definitions of key migration event (KME) states in a downsampled tree.**

| State of a KME | Parent node | Child node | ≥ 5 tips of the KME |
|---|---|---|---|
| Observed via internal nodes Observed via tips | Present | Present | Present |
| Erred via internal nodes Observed via tips | Present | Absent | Present |
| Obscured via internal nodes Observed via tips | Absent | Present or absent | Present |
| Observed via internal nodes Obscured via tips | Present | Present | Absent |
| Erred via internal nodes Obscured via tips | Present | Absent | Absent |
| Obscured via internal nodes Obscured via tips | Absent | Present or absent | Absent |

(at least 5) tips of the KME. The presence of the internal nodes of a KME indicates if we can correctly infer spatial-temporal information about the introduction (migration) event, and the presence of sufficient tips of the KME determines if there exists a large outbreak led by the introduction event. The reconstructed KME characterized by these two measures is summarized in Table 1 and explained below. The presence of internal nodes has three states: if both parent and child internal nodes of a KME appear in a downsampled tree, then we say the KME is "observed via internal nodes" in the downsampled tree; if only the parent internal node of a KME appears in a downsampled tree, then we say the KME is "erred via internal nodes" in the downsampled tree, and we measure the error by the branch length between the original child internal node of the KME and the apparent child internal node in the downsampled tree; if the parent internal node of a KME is not present in a downsampled tree, then we consider the spatial-temporal information of the introduction event can not be correctly reconstructed, and we say the KME is "obscured via internal nodes". The presence of sufficient tips of the KME simply has two states: if at least 5 tips of the KME appear in a downsampled tree, then we say the KME is "observed via tips"; if fewer than 5 tips of the KME appear in a downsampled tree, then we say the KME is "obscured via tips".

To assess the KME reconstruction accuracy for a true tree, we construct 100 downsampled trees for each fraction $k$ of location-A tips in downsampled trees; we count the number of downsampled trees in which each KME from the true tree is observed, erred and obscured via internal nodes and the number of downsampled trees in which each KME from the true tree is observed and obscured via tips.

## 2014–2015 Ebola epidemic

We illustrate the impacts of sampling differences using data from an Ebolavirus outbreak in Africa in 2014–2015 [32]. The data consists of 262 taxa collected from 4 countries: 152 from Guinea, 84 from Sierra Leone, 22 from Liberia, 2 from Mali, and 2 from unknown locations. The time-scaled tree is generated using BEAST with a log-normal distributed relaxed molecular clock and a non-parametric coalescent prior. More details can be found in the original paper [32].

Naturally, the published data only include some of the infections that occurred, but we can explore the impact of sampling by further downsampling, selecting among the tips that are present. We consider the reconstructed phylogeny with 262 tips as the true tree here; we reconstruct the location for the internal nodes of the tree with all 262 tips using ace and consider the reconstructed location of the internal nodes as true ancestral locations. Similarly, if a tip in the true tree has a different location as its parent node, then we say the tip is a "recent migrant". We apply two sampling schemes: randomly downsampling tips in one location at a time and preferentially downsampling recent migrants in the true tree. These tips are a proxy for travelers (and/or contacts of travelers). For random downsampling, since Liberia and Mali have

significantly fewer tips than Guinea and Sierra Leone in the phylogeny, we only consider downsampling Guinea and Sierra Leone tips. We downsample by dropping 80% of the tips in each of the two locations randomly. In addition, for the second downsampling scheme we also randomly downsample 80% recent migrants. Once we have a downsampled tree, we use ace to reconstruct the ancestral location for internal nodes included in the downsampled tree and compare reconstructed ancestral location to the true ancestral location (reconstructed with all 262 tips).

## Results

We consider the impact of the sampling scheme on the inference of two broad categories of phylogeographic inference: geography (the inference of ancestral states/locations) and migration (the inference of migration rates and key migration events). For each of these categories, we explore the effect of geographically biased sampling and sampling of tips for which there is recent migration in the true tree.

### Geography

**Absolute and relative reconstruction accuracy.** We find that the overall absolute accuracy of phylogeographic reconstruction is often high, particularly when the migration rate is low (Fig 1 and S1 Fig). Under a low migration rate, changing the fraction of the taxa that are from location A to half of what would be representative (from approximately 0.68 to 0.35 in Fig 1B) reduces the accuracy from being consistently above 95% to being in the range of 90–100%; while most nodes are correct, biased sampling greatly increases the number of errors. Under a low-to-moderate migration rate ($\alpha = 0.3$; Fig 1), halving the number of location-A samples in the tree reduces the average accuracy from above 90% to approximately 80%, more than doubling the number of internal nodes with an incorrect location estimate. Furthermore, the estimates are quite variable. Under higher migration rates, the accuracy is lower, but it is less sensitive to sampling.

In contrast to our initial expectation that absolute accuracy would peak under unbiased sampling (i.e. when the proportion of location-A tips in the downsampled tree was the same as the proportion in the true tree), we found that the absolute accuracy increased as the proportion of location-A tips in the downsampled tree increased (Fig 1B and 1D and S1 Fig). This is a result of the starting point that the root node is in location A, so that over half of the internal nodes are expected to be in location A (recall that migration from A to B and back are equal in our simulations). Without directly accounting for sampling bias in the ancestral state reconstruction, the absolute accuracy will therefore be higher when location A is oversampled. The higher absolute accuracy is simply because almost all of the tips and internal nodes of the downsampled tree are in location A, making it easy for the reconstruction to obtain correct locations. This is relevant particularly when the migration rate is relatively low. In this case, there are few migration events, and the true tree consists of larger monomorphic (same location) clades. This greatly improves the ancestral state reconstruction by increasing the probability that internal nodes have the same location as all their immediate descendants, and this raises the absolute accuracy.

In contrast to absolute accuracy, the relative accuracy (which takes this into account, and captures the improvement over expected accuracy from a null model; see Methods) peaks at a sampling proportion of 50% location-A tips, and the overall relative accuracy depends on the geographic sampling bias (Fig 1 and S1 Fig). When the migration rate is higher, the accuracy of the phylogeographic reconstruction is lower, and the dependence on sampling is less

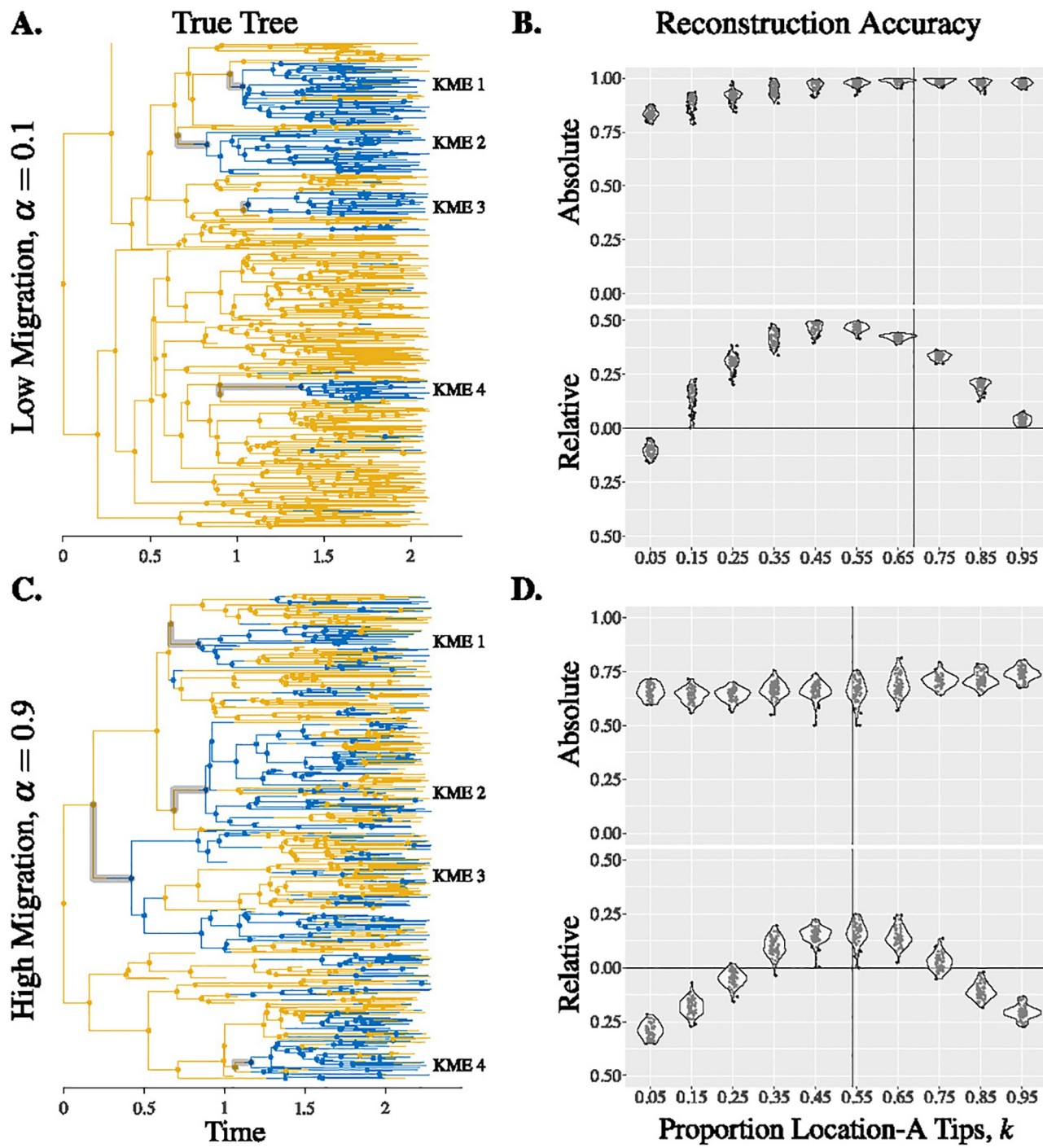

**Fig 1. Reconstruction accuracy depends on sampling bias and migration rate.** Panels A & C: True trees simulated with low or high migration rates, KMEs are highlighted. Panels B & D: Absolute and relative accuracy of downsampled trees. Each data point represents a downsampled tree from the true tree on the left. The vertical lines indicate the proportion of location-A tips in the true trees. Parameters: λ = 4 and μ = 1.

pronounced. Both the impact of sampling proportion and migration rate hold across a large number of true trees (S2 Fig).

The results in S3 Fig illustrates *how* the phylogenetic reconstructions are inaccurate. Reconstruction of migration events, or the lack thereof A2, between a parent and child node is

particularly sensitive to state of the parental node. If the parental state is over(under)-sampled, we are likely to greatly over(under)-represent those types of edges in the tree. In contrast, over- or under- sampling of the child state has a limited impact on geographic inference. This result is most pronounced when the migration rate is high. To interpret these results: suppose we are located in a region with comparatively high sequencing (location B; left side of the panels in S1 Fig). Sampling bias will impact our understanding of both migration between jurisdictions and the extent of sustained transmission within a jurisdiction. Specifically we will substantially underestimate the migration rate from the under-sampled jurisdiction (A-B edges) but fairly accurately capture migration to that jurisdiction from the highly sampled region (B-A edges). In addition, we are likely to conclude there are long sustained transmission chains in the over-sampled jurisdiction (B-B edges) and short transmission chains in the under-sampled jurisdiction (A-A edges).

We also explored how sensitive our results are to the assumption of equal 'speciation' rates. For example, transmission may be occurring at a higher rate in location A or B due to different public health measures or population contact rates. When transmission (or speciation, in the language of our model) is location-dependent, the overall patterns of bias are similar to what we have found in the neutral case. However, the maximum relative accuracy is obtained by sampling the location with the lower speciation rate more than would be representative of the locations' frequencies in the true tree (see S5 Fig).

**Oversampling recent migrants.** We find that both the absolute and the relative accuracy decrease as more recent migrants (recall that recent migrants are tips in a true tree that have a different location from its parent node) are sampled (Fig 2A and 2B and S6 Fig). We use recent migrants as models for either travelers or members of their contact networks. This means if we sample more travel-associated infections, mark the corresponding taxa with the location at which they were sampled, and use the taxa to infer ancestral states or migration patterns, then the inference accuracy may be reduced. We note (Fig 2C) that there exist tips with migration in their very recent ancestry in a downsampled tree that are not recent migrants in the true tree. We call such tips of the downsampled tree the "apparent" recent migrants. Fig 2D illustrates how the presence of both true and apparent recent migrants contribute to the inaccuracy in ancestral state reconstruction. The error caused by recent migrants or apparent recent migrants can cascade up the tree and affect the overall absolute and relative accuracy. For example, in Fig 2F on the right hand side, note the progression of error (red node) due to incorrect inference of location B instead of A, following on from including two tips with recent migration (rightmost tips in Fig 2E). If recent migrants are heavily sampled (Fig 2E and 2F), then the absolute accuracy of internal nodes can be much reduced.

## Migration

**Migration rate inference.** The maximum likelihood method that we use for ancestral state reconstruction gives estimates of migration rates assuming that migration is symmetric between the two locations. We find that at the least biased geographic sampling proportions, the estimates of migration rates are close to the migration rates of the true trees, but the best migration rate estimates do not always occur when the geographic sampling is least biased (Fig 3). Migration rates are on average overestimated when the true migration rate is low (Fig 3 top panels) whereas rate estimates are highly variable but on average closer to the truth when the true migration rate is high (Fig 3 bottom panels). Migration rate estimation is presumably less biased under higher migration because there are more migration events on which to base estimates. However, as the number of migration events increases, so too does the sensitivity to biased sampling (more migration events are incorrectly inferred). Note that the y-axis in Fig 3

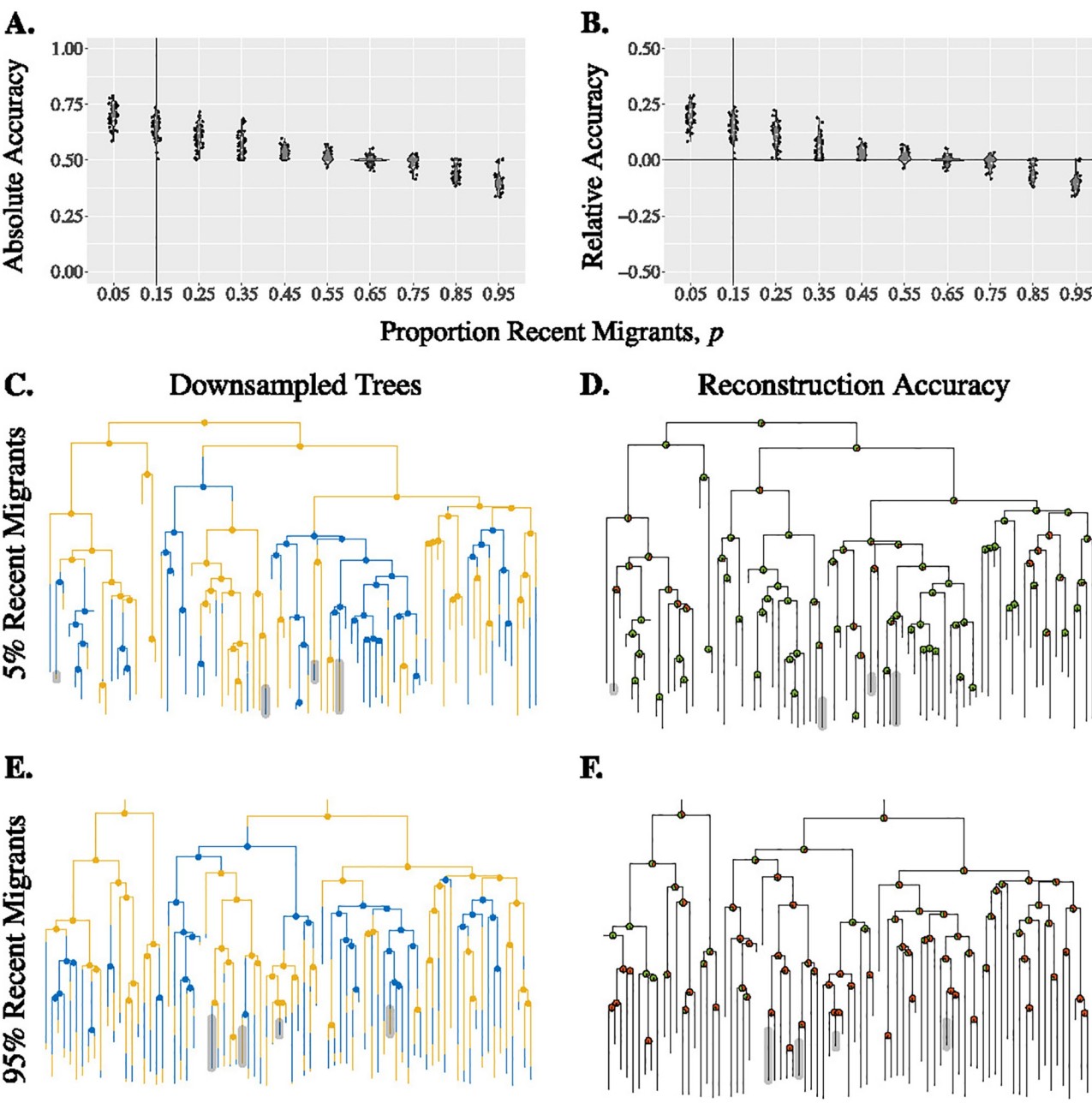

**Fig 2. Reconstruction accuracy depends on the sampling of recent migrants.** Absolute (Panel A) and relative (Panel B) accuracy of downsampled trees with varying $p$. The corresponding true tree is shown in Fig 1C with the vertical line showing the proportion of recent migrants in this true tree. Panels C & E: Examples of downsampled trees from the true tree in Fig 1C with a low (Panel C: $p = 0.05$; recent migrants highlighted) or high (Panel E: $p = 0.95$; non-migrants highlighted) proportion of recent migrants tips. Panels D & F: The absolute accuracy of each internal node (green fraction) of the downsampled trees in panels C and E respectively. Parameters: $\lambda = 4$, $\mu = 1$, $\alpha = 0.9$, and $k = 0.5$.

is on a logarithmic scale, the migration rate is often underestimated under extreme bias and overestimated when sampling is unbiased, especially when the true migration rate is high. When sampling is highly biased, the downsampled tree contains far fewer migration events than the true tree (clades become more monomorphic) which results in an underestimate of the migration rate. This is particularly true when the migration rate is high and there are more

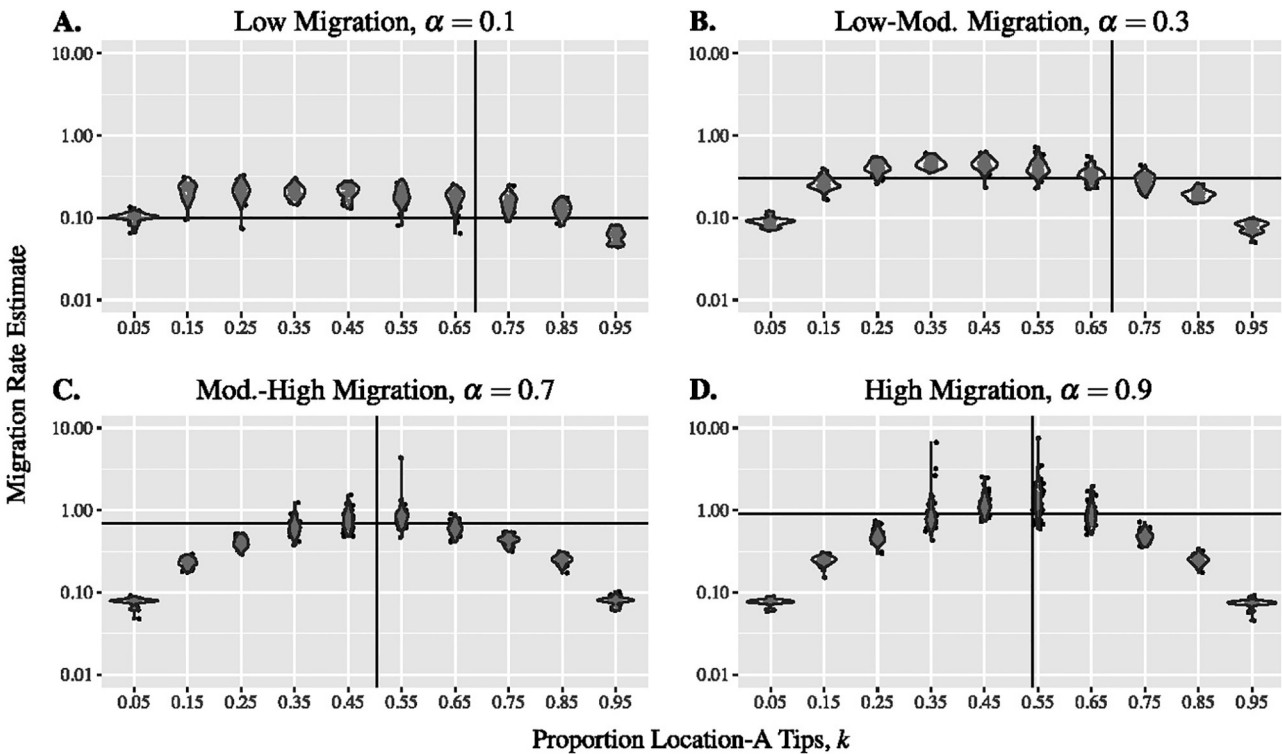

**Fig 3. Sampling bias affects the migration rate estimate.** Estimated migration rates from trees with varying migration rate. The true trees for each panel are shown in Fig 1A (Panel A), Fig 1C (Panel B),Fig 4A (Panel C), and Fig 1C (Panel D). Vertical lines indicate the true proportion of location-A tips and the horizontal lines indicate the true migrations rates. Parameters: $\lambda = 4$ and $\mu = 1$.

migration events that can be excluded by biased sampling. In contrast, the overestimation of migration rate when sampling is unbiased results from the fact that downsampling makes clades less monomorphic than in the true tree.

**Key migration events.** Sampling bias impacts whether and how key migration events are reconstructed. This also depends on the migration rate. Overall, we find that biased sampling can have a substantial impact on detecting migrations and can lead to either missing events entirely, or constructing them incorrectly. Recall that a KME is an edge in the true tree with a location-A parent internal node a location-B child internal node, and the subsequent location-B tips of the child node of the KME are the tips of the KME. Whether the edge can be reconstructed in a downsampled tree indicates whether we can learn about the introduction event from the sample. Whether we sample sufficient location-B tips impacts whether we can accurately infer that the introduction led to onward transmission in the destination. The number of KMEs that are "observed via tips" decreases as the number of location-A tips in the downsampled tree increases (Fig 4 and S7 Fig) because, if we do not sample enough location-B tips, we will not know there is ongoing transmission after an introduction. Note that there are always two child lineages arising from the parental node of a KME: the focal child node which is in location B and a sister clade which is, at least initially, in location A. For a KME to be "observed via internal nodes" requires that the parental node is in the downsampled tree. This in turn, depends on whether tips in the sister clade of the KME is sampled (Fig 4 and S7 Fig).

The sampling policy most likely to reconstruct KMEs depends on the migration rate. When the migration rate is high, sampling more location-B tips helps reconstruct KMEs, since the true tree has few monomorphic clades and the KMEs would rarely be "erred" or "obscured via

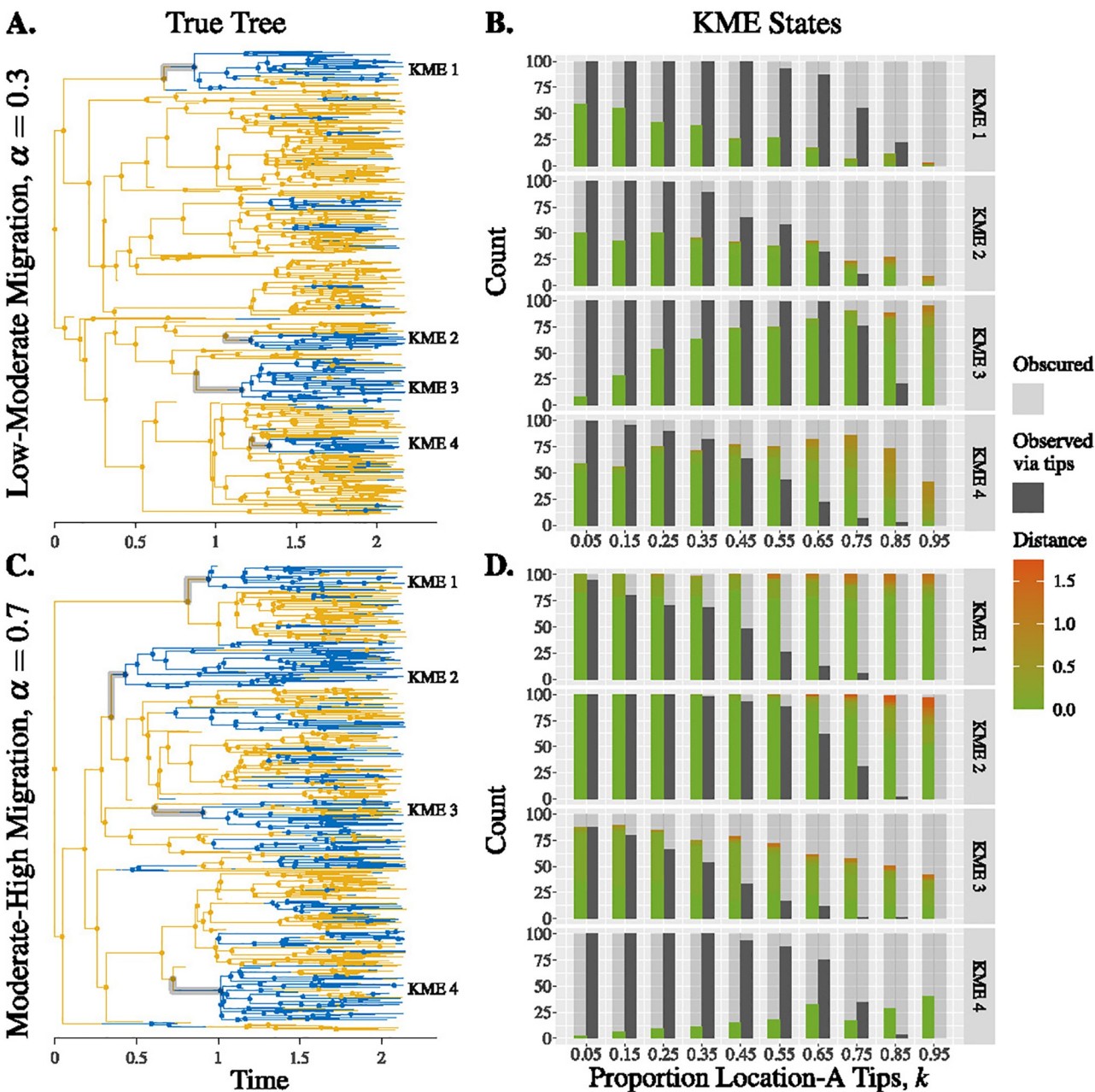

**Fig 4. Reconstruction of KMEs depends on sampling bias and migration rate.** Panels A & C: True trees with intermediate migration rates with key migration events highlighted. Panels B & D: The KME states (see Table 2) for 100 downsampled trees with low-moderate migration rate. Number of downsampled trees in which a KME is erred (coloured bars) or observed via internal nodes (left-hand transparent-grey bars) and whether KMEs are observed via tips (solid-grey bars) or obscured via tips (right-hand transparent-grey bars); for erred KMEs, the colors indicate the distance (time difference) between the child node in the true tree and the child node in a downsampled tree. Parameters: $\lambda = 4$ and $\mu = 1$.

internal nodes". When the migration rate is low, unbiased sampling gives better results because the true tree has monomorphic clades and heavily biased sampling can drop entire clades and cause the KMEs to be erred or obscured. Biased sampling comes at an additional cost to reconstruction of KMEs. If sampling is biased enough such that both parent and child internal nodes of a KME are reconstructed with the same location, then we may not identify

the KME even though both parent and child internal nodes and the sufficient number of tips of the KME appear in the downsampled tree (S5 Fig).

## Application to the 2014–2015 Ebola epidemic

We apply our analysis to the phylogenetic tree reconstructed from sequences of Ebola virus. The sequences are sampled at five locations, but most of them are from two locations, namely Guinea and Sierra Leone. The reconstructed maximum clade credibility (MCC) phylogenetic tree from BEAST is used as the "true tree", which has an estimated migration rate of 0.40 per unit time, from ace under the assumption of neutral migration rates between locations.

Relative to the simulations, the Ebolavirus tree has a relatively low migration rate compared to the branching rates, resulting in large monomorphic clades. We would therefore expect the absolute accuracy to be high overall, in keeping with Fig 1, but that the relative accuracy would depend on the geographic sampling bias. We examine the absolute accuracy of the internal nodes in downsampled trees, especially the internal nodes representing an introduction event in the true tree. Downsampling 80% of Sierra Leone tips does not greatly reduce the absolute accuracy of the downsampled tree (see S9 Fig). This is because the Sierra Leone tips form a monomorphic clade. When downsampling 80% of Guinea tips (less arranged in monomorphic clades), the overall absolute accuracy of the internal nodes is also high (see S10 Fig), but we observe a group of internal nodes (gray box in S10 Fig) that have inaccurate reconstructed locations. This does not have a large impact on overall absolute accuracy, but it significantly affects our inferences about the introduction event. Specifically, in this case we would infer that Ebolavirus was introduced to Guinea via Liberia, which is not the case in the true tree.

When we downsample 80% of the "recent migrant" tips (reflecting more likely recent-travel-related sequences) in addition to dropping 80% of tips independent of location, the ancestral state reconstruction is accurate except at the migration event in the gray box (Fig 5). This is because the reconstructed MCC tree has relatively few recent migrants (32/262), and removing most of these tips does not affect the absolute accuracy. However, if we were particularly interested in the information in the gray box in Fig 5, that is, whether the geographic spread is directly from Sierra Leone to Guinea or from Sierra Leone to Liberia to Guinea, then the impact of sampling travelers (or those with recent travel contact) would be of concern. Table 2 summarises A5 how inference of the likely source (parent location) and destination (child location) of introductions is impacted by down-sampling of recent migrants. After downsampling, the introduction event appears to be most likely from Sierra Leone to Guinea, whereas the introduction event in the true tree is from Liberia to Guinea.

## Discussion

We performed a simulation study to characterize the impact of geographic sampling bias on reconstructions of past viral locations, and to investigate how and why sampling bias affects the ancestral state reconstruction. Downsampling—and sampling bias in particular—can create two types of error in phylogeographic inference. First, the proportion of tips in each location can affect the overall accuracy of reconstructed ancestral locations. The error created near the tips can cascade up the tree, as node locations are estimated using their descendants' locations. Second, with biased sampling, the reconstructed tree shape can omit nodes and edges related to key migration events, making inference about important introduction events difficult. The eventual accuracy depends on the underlying migration pattern, the migration rate, the rate of sampling bias and the rate of sampling travelers. We found that the relative accuracy is worsened with increased sampling bias and when travel-associated tips are over-represented. If the underlying migration rate is higher, the relative accuracy of phylogeographic

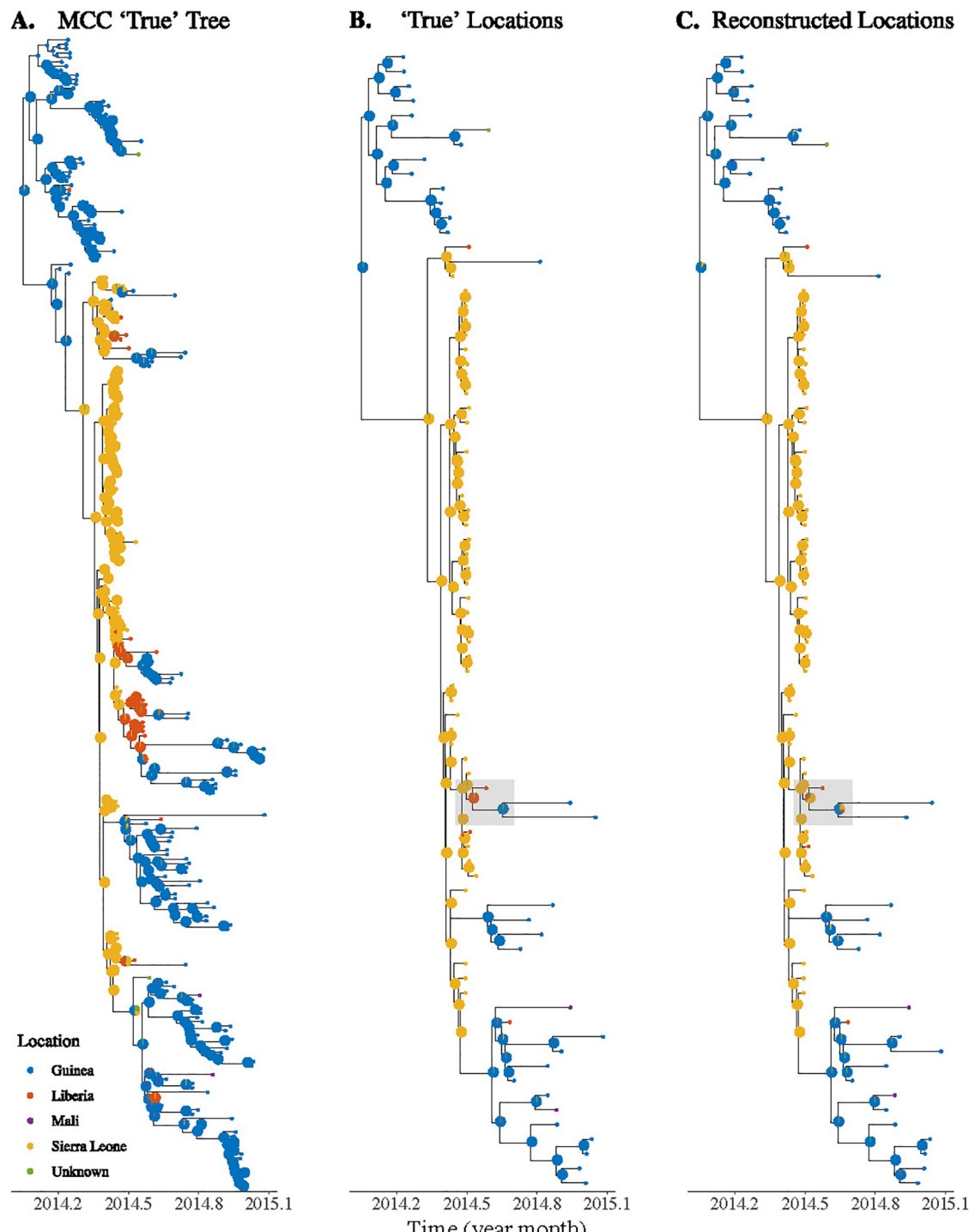

**Fig 5. Ancestral state reconstruction of Ebolavirus trees with recent migrants downsampled.** Panel A: The Maximum Clade Credibility (MCC) tree of Ebolavirus sequences generated using BEAST, which is used as the true tree. Downsampled trees obtained by dropping 80% recent migrants with either the true ancestral locations (Panel A) or ancestral locations reconstructed after downsmapling migrants (Panel C).

**Table 2. Effect of downsampling on inferred introduction events.** Likelihood of the parent node and the child node of a migration event in the gray box being at each location before and after downsampling recent migrants.

| Location | Guinea | Liberia | Mali | Sierra Leone | Unknown |
|---|---|---|---|---|---|
| Parent node (before) | 0.00 | 0.99 | 0.00 | 0.01 | 0.00 |
| Child node (before) | 0.99 | 0.01 | 0.00 | 0.00 | 0.00 |
| Parent node (after) | 0.08 | 0.29 | 0.01 | 0.61 | 0.01 |
| Child node (after) | 0.62 | 0.12 | 0.02 | 0.22 | 0.02 |

reconstructions after sampling can be lower, but inferences about migration rates and about key migration events can be more accurate than if the underlying migration rate is lower. While we find that biased sampling impacts ancestral state reconstruction, reconstruction accuracy remains moderately high, and very high when migration rates are low. However, sampling bias impacts the inference of key migration events and the inference of the origins of transmission chains. Even with a high overall accuracy, changes in sampling can double the number of incorrect ancestral locations, with a high impact on the reconstruction of viral movements. Where migration rates are higher, sampling has a lower impact, but the reconstruction quality is lower as well. Bias arising, for example, from differences in surveillance intensity leads to the phylogeographic bias towards locations with high sampling rates.

We only simulated two locations in our model, and in the main text we used a birth-death model without location-depending speciation rate or migration rate. We show in the (S1 Text) that location-dependant speciation rate can also impact phylogeographic reconstruction, and in this case oversampling the location with a lower speciation rate (compared to that location's representation in the true tree) can help to compensate. We did not vary the size of downsampled trees, and we assumed that a true tree is large enough to contain both at least $n$ location-A tips and at least $n$ location-B tips, which may be an unrealistic assumption. We chose to generate true trees first and simulate geographic sampling by downsampling the true trees after they are generated. However, we note that there are methods to simulate geographic sampling along a birth-death process, for example in [33]. We compared these two approaches to simulate geographic sampling and construct downsampled trees in the (S1 Text), and we found the two approaches provide similar results in terms of absolute reconstruction accuracy, hence relative reconstruction accuracy (see S11 Fig).

In the phylogenetic tree reconstructed from sequences of Ebolavirus, the true locations of the virus in the past are not known, and we used the reconstructed ancestral states from ace as the 'true' ancestral states. The reconstructed phylogenetic tree has a relatively small migration rate and monomorphic clades. Dropping tips from one location does not much affect the overall absolute accuracy, but can significantly reduce the quality of inference about introduction events. Similarly, the phylogenetic tree does not contain many recent migrants, and dropping most of the recent migrants does not reduce the absolute accuracy by much but can heavily affect inference about introduction events.

Furthermore, we simulated trees and either downsampled those trees or simulated birth, death and sampling through time; we did not simulate sequences evolving on the trees, nor reconstruct trees from these simulated sequences. Accordingly, our results are for the idealized circumstance where the reconstruction of the tree itself is essentially perfect. In practice, tree uncertainty would add additional noise to phylogeographic reconstruction, and the accuracy of reconstructions would be expected to be lower than we have found here.

We have found that even under idealized circumstances, the portion of infections from the different locations that are sequenced and represented in the data, and the extent to which

they are representative samples of the circulating infections in their jurisdictions, can strongly shape the reliability of phylogeographic inferences. While this was known in its impact on estimates of migration rates and the originating sequence (root), the impact on the estimated virus locations themselves, or on the ability to detect important viral movements, has not previously been characterized. Methods are being developed to compensate for this bias, but currently, these are Bayesian methods with high computational costs, and they are A6 not suitable for the analysis of the high volumes of viral sequences that are being generated. Furthermore, while incorporating past locations into Bayesian phylogenetic reconstruction is appealing, the fact that the internal nodes in the posterior sample of phylogenetic trees are not the same from tree to tree hinders interpretation of the reconstructed locations. This analysis is therefore more suited to estimating rates than viral movements. One approach is to proceed as in Lemey *et al.* [8], summarizing the posterior trees to capture viral movements without focusing on individuals nodes (though this does not resolve the issue of computational capacity for Bayesian tree reconstruction beyond hundreds of sequences). Another would be to use the BASTA framework of de Maio *et al.* [23], but if there are too many sequences for Bayesian phylogenetic estimation to be practical, fix the phylogenetic tree (estimating it first by maximum likelihood), and take the structured coalescent approach to the phylogeographic reconstruction. This may be limited by the assumptions of the structured coalescent (like fixed effective population sizes for the locations, or demes).

Adjustment for sampling differences is likely to be feasible only when those differences are known. Sharing data on the fraction of cases that are sampled and sequenced, the fraction of infections that are reported as cases, and the strategy by which samples are prioritized for sequencing, will help in making these adjustments. Ultimately this will aid in correctly inferring pathogens' geographic movements. Our results suggest that pathogens are likely to be overly estimated in jurisdictions that contribute more data (compared to the numbers of infections), and that over-representation of travel-associated samples can drive inaccurate estimates of past locations, incorrectly placing nodes in the location of sampling. If the reason for sequencing (e.g. travel-associated case; dense sampling due to a large outbreak) were known, this could be reduced. If the relative sampling fractions were known, a representative sample could be obtained by downsampling taxa from the relevant jurisdictions (though developing methods that can use all of the data is to be preferred). Meanwhile, we would caution that in real datasets from public databases, covering multiple jurisdictions sequencing viruses at potentially very different rates and in non-representative ways (due to targeting outbreaks, travellers, health-care workers, specific variants and so on for prioritized sequencing but then not making the reason for sequencing available), the results of phylogeographic reconstructions should be taken with caution.

## Supporting information

**S1 Text. Supplementary text.**
(PDF)

**S1 Fig. Reconstruction accuracy for intermediate migration rates.**
(TIFF)

**S2 Fig. Reconstruction accuracy across true trees.**
(TIFF)

**S3 Fig. Characterization of tree edges with low versus high migration rates.**
(TIFF)

**S4 Fig. Characterization of tree edges with intermediate migration rates.**
(TIFF)

**S5 Fig. Reconstruction accuracy with location-dependent speciation.**
(TIFF)

**S6 Fig. Effect of over/under-sampling recent migrants with different proportions of location-A tips.**
(TIFF)

**S7 Fig. Types and counts of KMEs.**
(TIFF)

**S8 Fig. Reconstruction accuracy of KME.**
(TIFF)

**S9 Fig. Ancestral state reconstruction of Ebolavirus trees with Sierra Leone downsampled.**
(TIFF)

**S10 Fig. Ancestral state reconstruction of Ebolavirus trees with Guinea downsampled.**
(TIFF)

**S11 Fig. Reconstruction accuracy of trees of extinct lineages trees versus trees simulated from a birth-death-sampling process.**
(TIFF)

**S12 Fig. Accuracy of root reconstruction depends on tree shape.**
(TIFF)

**S13 Fig. Accuracy of root reconstruction depends on migration rate.**
(TIFF)

## Author Contributions

**Conceptualization:** Caroline Colijn, Ailene MacPherson.

**Formal analysis:** Yexuan Song.

**Funding acquisition:** Caroline Colijn.

**Methodology:** Pengyu Liu.

**Software:** Pengyu Liu.

**Supervision:** Caroline Colijn, Ailene MacPherson.

**Visualization:** Pengyu Liu, Yexuan Song.

**Writing – original draft:** Pengyu Liu, Yexuan Song.

**Writing – review & editing:** Caroline Colijn, Ailene MacPherson.

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
