## [Decision Letter · Decision Letter 0]

9 Aug 2022

PGPH-D-22-00796

The impact of sampling bias on viral  phylogeographic reconstruction

Dear Dr. MacPherson,

Thank you for submitting your manuscript to PLOS Global Public Health. After careful consideration, we feel that it has merit but does not fully meet PLOS Global Public Health’s publication criteria as it currently stands. Therefore, we invite you to submit a revised version of the manuscript that addresses the points raised during the review process.

We look forward to receiving your revised manuscript.

Kind regards,

Everton Falcão de Oliveira, Ph.D

Academic Editor

Journal Requirements:

1. Please amend your detailed online Financial Disclosure statement. This is published with the article. It must therefore be completed in full sentences and contain the exact wording you wish to be published.

Please state what role the funders took in the study. If the funders had no role in your study, please state: “The funders had no role in study design, data collection and analysis, decision to publish, or preparation of the manuscript.”

2. Please update your online Competing Interests statement. If you have no competing interests to declare, please state: “The authors have declared that no competing interests exist.”

3. Please ensure that you provide a single, cohesive .tex source file for your LaTeX revision. You may upload this file as the item type 'LaTeX Source File.' As stated in the PLOS template, your references should be included in your .tex file (not submitted separately as .bib or .bbl). Please also ensure that you are making any formatting changes to both your .tex file and the PDF of your manuscript. If you have any questions, please contact Latex@plos.org. You can find our LaTeX guidelines here: https://journals.plos.org/globalpublichealth/s/latex

4. Please provide separate figure files in .tif or .eps format and ensure that all files are under our size limit of 10MB. If you are using LaTeX, you do not need to remove embedded figures.

5. We notice that your supplementary figures are included in the manuscript file. Please remove them and upload them with the file type 'Supporting Information'. Please ensure that each Supporting Information file has a legend listed in the manuscript after the references list.

Reviewers' comments:

Reviewer's Responses to Questions

**Comments to the Author**

1. Does this manuscript meet PLOS Global Public Health’s publication criteria? Is the manuscript technically sound, and do the data support the conclusions? The manuscript must describe methodologically and ethically rigorous research with conclusions that are appropriately drawn based on the data presented.

Reviewer #1: Yes

Reviewer #2: Yes

2. Has the statistical analysis been performed appropriately and rigorously?

Reviewer #1: Yes

Reviewer #2: Yes

3. Have the authors made all data underlying the findings in their manuscript fully available (please refer to the Data Availability Statement at the start of the manuscript PDF file)?

Reviewer #1: Yes

Reviewer #2: Yes

4. Is the manuscript presented in an intelligible fashion and written in standard English?

Reviewer #1: Yes

Reviewer #2: Yes

5. Review Comments to the Author

Reviewer #1: In this study, the authors investigate the potential bias on phylogeographic inference of viral outbreaks and transmissions introduced through the sampling process. In general, the manuscript is well and clearly written, analyses and deduced inferences appear sound and the figures suit content and results. Therefore, I would recommend this manuscript for publication in PLOS Global Public Health, however, I would also recommend the following few minor corrections prior to publication:

M & M:

Despite mentioning the source of the data in the data availability statement, I would suggest mentioning the source of the seqeunce data which subsequent analyses are based on at the beginning of the M & M section, otherwise the reader is unsure about whether the sequences have been produced in this study or retreived from an online repository.

line 218: '...the there..', please resolve

line 305: delete space between there and of

line 337 and onwards: there's something wrong with this sentence, please correcct/rephrase

Fig4 legend: low-moderate what? I think there is something missing here. It's 'whether' here not 'weather', right?

line 425: summarises

line 486: delete one 'are'

line 505: delte one 'to be'

Supplement:

page 26: 'Here we simulate under with...' appears grammatically odd to me.

Fig S5: 'Reconstruction accuracy for with...' appears grammatically odd to me.

Other than that, I congratulate the authors to a very nice manuscript. If questions arise through my comments, I would be happy to be of further assistance.

Best regards

Reviewer #2: The manuscript provides an important insight into sampling bias in phylogeographic trees. It's well written, concise and contributes substantially to phylogeographic studies, drawing the attention of the scientific community to the impact that sampling bias can have on the analyzes carried out.

The manuscript follows the journal's guidelines, the discussion was well conducted and the authors use other articles published in Plos journals. The figures are in the required formats. There are no more considerations to be made.

I just suggest that the color palette in figures 5, S9 and S10 be changed, allowing for a better distinction between the locations.

6. PLOS authors have the option to publish the peer review history of their article (what does this mean?). If published, this will include your full peer review and any attached files.

**Do you want your identity to be public for this peer review?** For information about this choice, including consent withdrawal, please see our Privacy Policy.

Reviewer #1: No

Reviewer #2: No

---

## [Editor Report · Decision Letter 1]

19 Aug 2022

The impact of sampling bias on viral  phylogeographic reconstruction

PGPH-D-22-00796R1

Dear Dr. MacPherson,

We are pleased to inform you that your manuscript 'The impact of sampling bias on viral  phylogeographic reconstruction' has been provisionally accepted for publication in PLOS Global Public Health.

Best regards,

Everton Falcão de Oliveira, Ph.D

Academic Editor
